# Research on the relationship between virtual social interaction and the degree of loneliness based on algorithm matching technologies: A quantitative analysis on the SOUL APP-A virtual social software for strangers

Linsha Liu👁* 

School of Journalism and Communication, Chengdu Sport University, Chengdu, Sichuan, P.R. China

* liulinsha@swjtu.edu.cn

## Abstract

This study examines the relationship between virtual social interaction and people's social behaviors and psychology using algorithm matching technologies and questionnaire surveys. The focus is on interpersonal communication on virtual social platforms. The findings indicate that engaging in virtual social networking is often accompanied by a high level of loneliness. Users who experience social anxiety in the real world tend to feel more lonely, and this loneliness is exacerbated by presenting an unreal version of oneself and having distrust in virtual social networking. Users with higher anxiety and loneliness levels are more likely to use the algorithm matching function of virtual social networking, engage in false self-presentation, and have less trust in the platform. Since the inherent flaws of virtual social networking cannot be eliminated solely through algorithm matching, a potential solution is to introduce more offline to online social functions for strangers. This exploration of actual matching on social platforms may help reduce users' loneliness.

## Introduction

In recent years, the rapid advancement of media technologies has ushered in a new era of digital and smart media. Through the application of big data and intelligent technologies, human cognitive processes, motor skills, and even cultural preferences are increasingly being intelligently matched and synchronized. As a result, audiences gain a stronger sense of immersion and a more intuitive online experience. According to data from QuestMobile [1], the number of mobile networking users in China reached 1.224 billion in 2023. Since 2018, several new social networking apps have been launched, including popular apps like Zhiya, Soul, and Yinyu. These apps strive to create enriched and diversified social activities while improving platform matching efficiency and user experience through the application of algorithms, big data, and AI (Artificial Intelligence).

**Data Availability Statement:** All relevant data are within the paper and its Supporting information files.

**Funding:** The author(s) received no specific funding for this work.

**Competing interests:** The authors have declared that no competing interests exist.

This study takes the Soul App as a case to explore the impact of AI-driven virtual social behaviors on individual loneliness. It aims to uncover how the authenticity of self-presentation in virtual interactions, the sense of trust in these environments, the pressures of real-world social interactions, and loneliness interact to create a cyclical effect. Additionally, the study seeks to evaluate the effectiveness of AI matching technology in enhancing authenticity and trust in virtual social interactions, and whether these improvements can alleviate users' feelings of loneliness. The study ultimately aims to provide insights into the effects of virtual social interactions on individual mental health, offering valuable guidance for the development and design of virtual technologies.

## Part One: Research background

"Through technology, we manifest and extend our thoughts, injecting them into the objective world and shaping it according to our designs," Paul Levinson emphasizes human agency [2]. He argues that in the evolution of media, rational choices made by humans can address both the shortcomings in media development and the limitations inherent in media themselves. Media technologies, as they evolve, increasingly tend to meet human needs and facilitate the exchange of information [3]. Each new medium is a remedy for its predecessors, making media more and more suited to human needs. "Why We Expect More from Technology and Less from Each Other?" This is the question posed by Sherry Turkle, about the influence of internet technology on human interactions in the real world [4]. Supported by network technology, people can maintain constant contact with others beyond the constraints of time and space, yet at the same time, they ignore those who are physically near. With the development of network technology, people seem to have become more vulnerable; the more noise there is in the virtual world, the more loneliness they experience in reality. Sherry Turkle's views have resonated widely with the public, and the term 'Alone Together' has become a reflection of the psychological state of people in the networked society. Both Levinson and Turkle believe that the development of media technology has a significant impact on humanity, yet their conclusions are completely opposite. Levinson argues that the evolution of media adapts to human needs, while Turkle believes that technological development inevitably leads to human alienation. In the digital age, how does the issue of 'Alone Together' differ from what Sherry Turkle observed in her 2011 study? As technology achieves higher levels of development, does new media technology accommodate human needs, or does it further alienate humanity?

This study is based on the theoretical framework of Media Ecology and aims to investigate the dynamic influences of the co-existence of media technologies and humans. It also explores the potential of a synthetic symbolic context that emerges through their mutual interactions. The study focuses on studying the communicative and ecological systems shaped by media and the media context. The goal is to uncover the mutual interactions between media technologies, humans, and social context, and to explore possible approaches for achieving a more balanced communicative context system.

## Part Two: Literature reviews & hypotheses

### I. Realistic dilemma: Research on the relationship between real-world social anxiety of virtual social users and the degree of loneliness

According to William B, human beings nowadays tend to prefer engaging with the virtual world rather than experiencing the tangible reality outside their windows [5]. This preference has become more evident with the advancement of virtual technologies. As the virtual world becomes more realistic, individuals who are addicted to virtual social networking find it

increasingly challenging to willingly engage with the real world. This raises the question: how does the user's virtual behavior impact their actual situation? Specifically, what are the relationships between the user's virtual social behavior, degree of loneliness, and social anxiety?

Social anxiety, a common problem in interpersonal social interactions [6], is characterized by emotional experiences such as fear, tension, and anxiety during communication [7]. Individuals with social anxiety fear interacting with strangers and worry about how they are perceived, often feeling embarrassed and ashamed of their fear and nervousness in social situations [8]. Loneliness, on the other hand, refers to the experience of emotional and social isolation, highlighting the subjective nature of this feeling [9]. Russell et al. developed a measure of loneliness to quantify this subjective experience, emphasizing the need for standardized tools to assess loneliness [10]. Moore et al. focused on loneliness in adolescence, examining its correlates, attributions, and coping mechanisms among young individuals [11].

Numerous studies have shown a strong positive correlation between loneliness and social anxiety in both youth and adulthood [12]. Research by Sun Mengyuan and Liu Kun demonstrated that social anxiety significantly predicts loneliness in middle school students [13]. In older adults, Sun S et al. further emphasized that social anxiety strongly predicts loneliness, with perceived social support moderating this effect [14]. This establishes the foundation for exploring the relationship between loneliness in virtual social interactions and real-world social anxiety, indicating that anxiety and loneliness are mutually reinforcing. Additionally, Enez et al. found a positive correlation between smartphone addiction, social anxiety, and loneliness among university students in Istanbul [15]. Zhou X et al., using a network approach, revealed that loneliness and escapism mediate the relationship between social anxiety and social comfort [16]. Based on this, we deduce that frequent use of smartphones or virtual social platforms may exacerbate loneliness, especially when users are trying to escape real-world social pressures. Therefore, we propose the first hypothesis:

H1: The actual social anxiety of virtual social users positively predicts their degree of loneliness.

## II. Internet disadvantages: Studies on the relationship between the real-world social anxiety, degree of loneliness and virtual social behavior

The relationship between the utilization of social networking and the level of loneliness experienced by users has not been clearly established in current studies, leading to different perspectives. For example, Burke, Marlow, and Lento argue that the use of social networking is positively correlated with loneliness [17], while Teppers, Luyckx, Klimstra, and Goossens hold the view that frequent utilization of social networking reduces loneliness [18]. In this regard, YAO Qi, MA Huawei, YAN Huan, and CHEN Qi suggest that future research should focus on specific behaviors within social networking [19]. Existing studies on virtual social behavior often examine the use of virtual self-presentation and virtual social trust as two types of social networking.

Self-presentation, also known as self-expression and impression management, was introduced by Goffman E [20]. It refers to the effort individuals make to shape the way others perceive them and to create a desired self-image. The association between self-presentation and social anxiety has been a focal point in numerous research studies. Maddux et al. explored the combination of self-efficacy theory and self-presentation theory in understanding social anxiety, with a focus on the connection between general social anxiousness and expectations regarding anxiety-inducing social situations [21]. Martin et al. emphasized the significance of physical self-presentation in the context of trait anxiety during sports competitions, highlighting the relationship between fear of negative evaluation and competitive trait anxiety [22]. Gammage et al. investigated the cognitive aspects of self-presentation in exercise settings,

examining variables such as social physique anxiety, self-presentational efficacy, impression motivation, and exercise imagery among female exercisers [23]. Moreover, investigations conducted by Jankauskienė et al. analyzed the association between social anxiety and problematic internet use, exercise adherence, body image, and social physique anxiety [24]. These studies highlight the influence of self-presentation on individuals' behaviors and psychological well-being in various contexts. Recent research by Kehayes et al. delved into the connection between perfectionistic self-presentation and social anxiety, emphasizing the mediating roles of self-presentation motivation, expectancies, and anger rumination [25]. These studies collectively indicate that self-presentation concerns have a substantial impact on behavior and anxiety levels in various social and competitive environments.

The literature review on the association between self-presentation and social anxiety demonstrates a multifaceted link across various contexts such as social interactions, competitive settings, exercise environments, and online platforms. The studies discussed underscore the importance of considering self-presentation concerns in comprehending social anxiety and its implications for individuals' behaviors and psychological well-being. Thus, this study proposes a second hypothesis to explore this relationship.

H2: The real self-presentation of virtual social users is negatively predicted by their real-world social anxiety and negatively predicted the degree of loneliness.

H2.1: The real-world social anxiety of virtual social users negatively predicts the self-presentation of virtual social networking.

H2.2: The self-presentation of virtual social networking negatively predicts the degree of loneliness.

Previous studies have shown that social networking users with higher levels of social trust are more likely to engage in online communication and seek social interactions [26, 27]. Specifically, in social media apps, higher levels of social trust are associated with increased time spent on online social activities [28]. Users with greater social trust are also more active and open to accepting new things, indicating the mediating role of social trust in online social networking [29]. This study aims to investigate the mediating role of virtual social trust in the relationship between social anxiety and the degree of loneliness. As a result, the third hypothesis is formulated as follows:

H3: The social trust of virtual social networking users is negatively predicted by their real-world social anxiety, and negatively predicted the degree of loneliness.

H3.1: The real-world social anxiety of virtual social networking users negatively predicts their virtual social trust.

H3.2: The virtual social trust of virtual social networking users negatively predicts their degree of loneliness.

This study constructs a model based on the aforementioned assumptions, as shown in Fig 1:

If the above assumptions were confirmed, what are the impacts of algorithm matching technologies on the trust/self-presentation, loneliness, and real-world social anxiety of virtual social networking users?

## III. The temptation of quick matching: Study on the impacts of virtual social acceptance with application of algorithm matching technologies

Matching plays a vital role in the rational allocation of resources in many areas, ranging from market operation to people's daily lives. In economics, the term matching theory is coined for

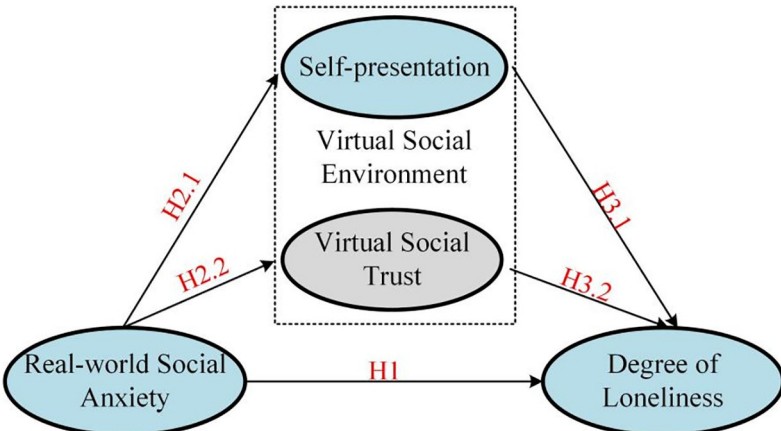

**Fig 1. Model diagram for virtual social behaviors, degree of loneliness, and social anxiety.**

pairing two agents in a specific market to reach a stable or optimal state. In computer science, all branches of matching problems have emerged, such as the question-answer matching in information retrieval, user-item matching in a recommender system, and entity-relation matching in the knowledge graph [30]. A matching algorithm is a type of algorithm used to dynamically compute similarity and synergy between different entities based on semantics and properties [31]. The combination of powerful computing, large amounts of data, and precise algorithms has facilitated the widespread use of algorithm matching technologies in social media platforms [32]. For example, in the case of the Soul App, after registering, users take a "Soul Assessment". They can also add "gravitational tags" to better showcase themselves. Using the results of the "Soul Assessment" and "gravitational tags," Soul's AI system analyzes users' interests and behavior to quickly recommend suitable, high-quality connections [33]. By utilizing artificial intelligence technology and targeted algorithms in this way to achieve personalized recommendations and improve the efficiency of matching relevant user information, this could create social interactions that better meet users' actual needs and reduce loneliness. Therefore, the fourth hypothesis is proposed:

H4: Virtual social networking users keen on algorithm matching technologies have a lower sense of loneliness

## IV. Looking for satisfied partners: Study on the relationship of algorithm matching technologies-driven society and virtual social interactions

ZHAO Jing and other scholars defined online social trust as the anticipation of conversations and commitments, both written and oral, by counterparts in interactive online social contacts involving risks [34]. Virtual social trust is manifested by being confidential with others' information during virtual interpersonal interactions while ensuring the authenticity of one's personal information. In essence, virtual social trust implies a virtual method for mutual identity in an online context. To investigate whether the accurate matching of algorithm technologies amplifies mutual identity and virtual social trust, the fifth hypothesis is as follows:

H5: Users keen on algorithm matching technologies have a better experience in virtual social interactions.

H5.1: Users keen on algorithm matching technologies are more willing to show self-presentation in virtual social interactions;

H5.2: Users ken on algorithm matching technologies have a higher sense of trust in virtual social interactions.

Individuals who struggle with socializing in real-world society may find online social activities to be time-consuming and draining. Wang Yiling discusses this inefficiency [35]. Fan Ziqing and Gao Yan analyze the emotional and psychological effects of online social activities on real-world social interactions [36]. Yang Siqing explores the challenges faced by college students in the digital age and highlights how virtual social activities can reduce their interpersonal skills in the real world [37]. Furthermore, with the rise of algorithm matching technologies in virtual social activities, individuals who have become accustomed to this process may encounter difficulties in engaging in real-world social interactions. This leads to the formulation of the sixth hypothesis:

H6: Users keen on algorithm matching technologies show more social anxiety in the real world.

Based on previous research, this study aims to summarize the key factors that influence the level of loneliness in real-world social activities. Additionally, it explores the impact of algorithm matching technologies on the degree of loneliness in virtual social interactions, considering factors such as factual self-presentation, interpersonal trust, and real-world social anxiety.

## Part Three: Design and methodologies of research

### I. Selection of samples

Soul App is a social networking platform application that emerged in 2015, aiming to alleviate users' sense of loneliness by introducing social partners who share many commonalities through data matching technologies. Additionally, data from 2023 revealed that Soul App has 9.55 million daily active users and has the highest average release rate and Z generation penetration rate among similar software in China [38]. Therefore, taking Soul App users as research objects has a good representation and can fully reveal the problems involved in this study.

Data collection for this study from Nov. 1, 2023 to Nov 20, 2023. The survey for this study was created using the online survey platform Wenjuanxing (www.wjx.cn, accessed on Nov. 1, 2023) and disseminated through Soul App. Among the 432 online collected questionnaires, 368 were deemed effective, resulting in an effective recovery ratio of 85.2%. For data analysis, SPSS 25.0 was used for descriptive analysis of the samples, while AMOS 25.0 was used for confirmatory factor analysis and hypothesis testing. The reliability test conducted revealed a Cronbach's Alpha coefficient of 0.768.

Ethical considerations were of utmost importance in this study. The study strictly adhered to ethical guidelines, which included obtaining informed consent, ensuring participant confidentiality, and protecting the rights and well-being of the participants. This study used anonymous online questionnaires to collect data and did not collect any personally identifiable information that could identify participants. At the beginning of filling out the online questionnaire, all potential participants were clearly informed of the purpose, methodology, potential risks and benefits of the study, and their voluntary participation was obtained. The questionnaire clearly informs participants that by submitting the questionnaire online, they are considered to have consented to participate in the study. The research protocol underwent

a thorough review and was approved by the Chengdu Sport University Research Ethics Board (2023#157).

## II. Variables measurement

**i. Anxiety on real-world social activities.** Referring to the Interpersonal Relationship Integrative Diagnostic Scale (IRIDS) developed by ZHENG R to measure real-world social abilities [39], this article evaluates the current situations of the participants' real-world social abilities, specifically focusing on their conversations and categories of friendship. The revised version of the scale includes five questions and use a five-point scoring system, with 1-point indicating a lack of match and 5-point indicating full compliance. The Internal Consistency Reliability value in this study is 0.898, and the KMO and Bartlett's Test of Sphericity values are 0.827 (p<0.001).

**ii. The scale of the degree of loneliness.** Based on a simplified version of the Loneliness Scale [40] and referring to the mental health assessment scale developed by Wang Xiangdong et al. [41] which is suitable for Chinese individuals (currently, there is no mature loneliness scale specifically based on the Chinese population), this study establishes four questions to assess the degree of loneliness experienced by the participants. The degree of loneliness is measured on a five-grade scoring system, with a higher score indicating a higher degree of loneliness. The Internal Consistency Reliability value in this study is 0.832, and the KMO and Bartlett's Test of Sphericity values are 0.693 (p<0.001).

**iii. Real self-presentation on virtual social.** Learning from the positive self-presentation and questionnaires developed by Kim J, Lee JE [42], and co-translated by NIU Gengfeng [43], this study applies a six-question scale to measure the extent of factual self-presentation on virtual social platforms. A five-grade scoring system is employed, where 1-point indicates no match and 5-point indicates full compliance. The scale demonstrates good reliability with an Internal Consistency Reliability value of 0.907, and the KMO and Bartlett's Test of Sphericity values are 0.887 (p<0.001).

**iv. Virtual social interpersonal trust.** The Online Interpersonal Trust Scale developed by DING Daoqun [44] and the Online Interpersonal Trust Scale created by NIU Jing, MENG Xiaoxiao [45] were utilized to assess virtual social interpersonal trust. This section consists of five questions to measure online interpersonal trust in the Soul App. And it is scored on a five-point scale, where 1 indicates no trust and 5 indicates complete trust. A higher score indicates a higher level of virtual trust, and vice versa. The Internal Consistency Reliability alpha coefficient for this scale was found to be 0.908, and the KMO and Bartlett's Test of Sphericity yielded values of 0.803 (p < 0.001).

## Part Four: Results

### I. Information for reference

Table 1 shows the demographic characteristics of the participants. The proportion of male participants is 44.8%, while female participants make up 55.2%. 77.9% of the respondents are under 24 years old, 17.11% are between 25 and 30 years old, 1.9% are between 31 and 40 years old, and those over 40 years old account for 2.99%. In terms of personal careers, college students represent the largest group at 51.63%, followed by company employees at 22.01%, while other occupations account for 26.36%. The monthly income of respondents shows that 40.76% earn below 3,000 RMB, 27.17% earn between 3,001 and 5,000 RMB, 17.93% earn between 5,001 and 10,000 RMB, and 14.13% earn above 10,000 RMB. The sample characteristics collected through the questionnaire survey are basically consistent with the user characteristics displayed by the Soul App [46], which indicates that the sample obtained from the

**Table 1. Demographic profile of the survey participants (N = 368).**

| Variables (N = 368) | Category | Frequency | Percent (%) |
|---|---|---|---|
| Gender | Male | 165 | 44.80% |
| | Female | 203 | 55.20% |
| Age | 18–24 | 287 | 77.90% |
| | 25–30 | 63 | 17.11% |
| | 31–40 | 7 | 1.90% |
| | 41–50 | 11 | 2.99% |
| Personal career | Entrepreneur | 6 | 1.63% |
| | Individual household | 1 | 0.27% |
| | Company management | 18 | 4.89% |
| | Company employees | 81 | 22.01% |
| | INTERN | 24 | 6.52% |
| | Freelance work | 46 | 12.50% |
| | multi-job | 2 | 0.54% |
| | College student | 190 | 51.63% |
| Monthly income | Less than ¥3000 | 150 | 40.76% |
| | ¥3001–¥5000 | 100 | 27.17% |
| | ¥5,001 -¥10000 | 66 | 17.93% |
| | More than ¥10,000 | 52 | 14.13% |

questionnaire has good representativeness and can effectively reflect the actual characteristics of Soul App users. This also demonstrates the reliability and validity of the research findings, enhancing the credibility of the study.

## II. Study on the relations of the variables

The relationships between real-world social anxiety, virtual social trust, self-presentation, and the degree of loneliness are presented in Table 2. Based on the statistics, the mean values for real-world social anxiety, virtual social trust, and self-presentation, as well as the degree of loneliness, exceed 3. The results indicate a positive correlation between virtual social trust and self-presentation (r = 0.658, p<0.001), as well as between the degree of loneliness and real-world social anxiety (r = 0.520, p<0.001). Conversely, real-world social anxiety is negatively associated with both virtual social trust and self-presentation. Specifically, real-world social

**Table 2. The correlation coefficient matrix, means, standard deviation, correlation coefficient, and square root of AVE (N = 368).**

| Item | Real-world social anxiety | Self-presentation | Virtual social trust | Loneliness |
|---|---|---|---|---|
| Real-world Social Anxiety | 0.807 | | | |
| Self-presentation | -0.766*** | 0.799 | | |
| Virtual Social Trust | -0.636*** | 0.658*** | 0.823 | |
| Loneliness | 0.520*** | -0.587** | -0.471*** | 0.762 |
| Mean | 3.122 | 2.810 | 2.667 | 3.196 |
| SD | 0.997 | 0.857 | 0.906 | 0.685 |

Notes:

*p<0.05,

**p<0.01,

***p<0.001; the value on diagonal represents the square root of AVE

**Table 3. Analysis of confirmatory factors (N = 368).**

| Latent Variable | Item | Parameter Estimate | | | | | | Convergence Validity | | |
|---|---|---|---|---|---|---|---|---|---|---|
| | | NSF | S.E. | t | p | SF | | SMC | CR | AVE |
| Real-world Social Anxiety | T1 | 1 | | | | 0.757 | | 0.573 | 0.903 | 0.651 |
| | T2 | 1.076 | 0.123 | 8.768 | *** | 0.775 | | 0.601 | | |
| | T3 | 1.011 | 0.12 | 8.419 | *** | 0.845 | | 0.714 | | |
| | T4 | 0.918 | 0.121 | 7.618 | *** | 0.876 | | 0.767 | | |
| | T5 | 0.971 | 0.128 | 7.598 | *** | 0.773 | | 0.598 | | |
| Self-presentation | T6 | 1 | | | | 0.935 | | 0.874 | 0.912 | 0.639 |
| | T7 | 1.878 | 0.372 | 5.054 | *** | 0.832 | | 0.692 | | |
| | T8 | 1.721 | 0.351 | 4.911 | *** | 0.86 | | 0.74 | | |
| | T9 | 1.972 | 0.382 | 5.156 | *** | 0.771 | | 0.594 | | |
| | T10 | 1.779 | 0.35 | 5.084 | *** | 0.821 | | 0.674 | | |
| | T11 | 2.112 | 0.397 | 5.323 | *** | 0.508 | | 0.258 | | |
| Virtual Social Trust | T12 | 1 | | | | 0.824 | | 0.679 | 0.912 | 0.677 |
| | T13 | 0.755 | 0.099 | 7.663 | *** | 0.72 | | 0.518 | | |
| | T14 | 0.884 | 0.098 | 9.007 | *** | 0.808 | | 0.653 | | |
| | T15 | 0.923 | 0.093 | 9.95 | *** | 0.865 | | 0.748 | | |
| | T16 | 1.015 | 0.098 | 10.313 | *** | 0.886 | | 0.785 | | |
| Loneliness | T17 | 1 | | | | 0.694 | | 0.482 | 0.84 | 0.581 |
| | T18 | 1.27 | 0.166 | 7.656 | *** | 0.897 | | 0.805 | | |
| | T19 | 1.249 | 0.163 | 7.651 | *** | 0.896 | | 0.803 | | |
| | T20 | 0.604 | 0.14 | 4.327 | *** | 0.482 | | 0.232 | | |

Notes:

*p<0.05,

**p<0.01,

***p<0.001.

anxiety is inversely related to virtual social trust (r = -0.636, p<0.001), and it is also negatively associated with self-presentation (r = -0.766, p<0.001). Similarly, the degree of loneliness is negatively associated with both virtual social trust and self-presentation. The degree of loneliness is inversely related to virtual social trust (r = -0.471, p<0.001), and it is also negatively associated with self-presentation (r = -0.587, p<0.01).

## III. Analysis of confirmatory factors

The figures in Table 3 indicate that the variables in the corresponding questions have a capacity exceeding 0.5 and fluctuate below p<0.001. Additionally, the Average Variance Extracted (AVE) of the variable exceeds 0.5, and the square root of AVE is greater than the correlation coefficient (as shown in Table 2). This suggests that the variables have good discriminatory validity [47]. Furthermore, the Composite Reliability (CR) value exceeding 0.8 indicates positive reliability of the variables in this study [48].

## IV. Tests of hypotheses

To investigate the relationship between real-world social anxiety, degree of loneliness, and virtual social trust and self-presentation, this study utilized AMOS25.0 to conduct SEM (Structure Equation Modelling). The aim was to examine the interrelated influences and analyze the direct effect, total effect, and mediation effect. The initial results were then revised and updated

**Table 4. Results of common fit indexes for SEM (N = 368).**

| Index | CMIN/DF | GFI | AGFI | CFI | RMESA | NFI | TLI | IFI |
|-------|---------|-----|------|-----|-------|-----|-----|-----|
| Value | 1.177 | 0.869 | 0.795 | 0.984 | 0.044 | 0.903 | 0.977 | 0.984 |

using Modification Indices. The fit of the model was assessed using Chi-square values, with a significance probability of p = 0.079>0.05, indicating a good fit. If the hypothesis is confirmed, it suggests that the observed figures align closely with the supposed modelling. Table 4 presents the findings, showing that the ratio of CMIN and DF of SEM is lower than 3, and the NFI, TLI, and IFI exceed 0.9, indicating a good fit. The RMSEA below 0.08 also demonstrates a good fit for SEM. Additionally, the CFI, GFI and AGFI exceed 0.7, indicating an acceptable extent of SEM fit.

**i. Tests of direct effect, total effect, and mediation effect.** Based on the data in Fig 2, it can be observed that real-world social anxiety has a positive impact on users' level of loneliness (β = 0.341, p<0.001). This means that individuals with higher levels of real-world social anxiety experience a greater degree of loneliness. Therefore, the first hypothesis (H1) is supported by relevant evidence. Additionally, real-world social anxiety has a negative effect on self-presentation (β = -0.354, p<0.001). This suggests that individuals with higher levels of real-world social anxiety tend to present themselves in an unrealistic manner, confirming H2.1. Furthermore, real-world social anxiety also has a negative influence on the user's virtual social trust (β = -0.588, p<0.001). Users with higher levels of real-world social anxiety exhibit lower levels of trust in virtual social interactions, confirming H3.1. The self-presentation in virtual social interactions also has a negative impact on the level of loneliness (β = -0.180, p<0.05). This implies that the more individuals engage in untrue self-presentation in virtual social settings, the higher their level of loneliness becomes, confirming H2.2. Similarly, virtual social trust has a negative effect on the level of loneliness (β = -0.317, p<0.001). This indicates that individuals who have more trust in virtual social interactions tend to experience less loneliness in real society, supporting H3.2. Overall, all the proposed hypotheses have been confirmed and have revealed a paradoxical relationship between the level of loneliness and real-world social anxiety. To mitigate loneliness, it is important to engage in more authentic self-presentation and develop trust in virtual social activities. However, due to anxiety and frustrations, individuals using virtual social platforms are often hesitant to present their true selves and have limited trust in such activities.

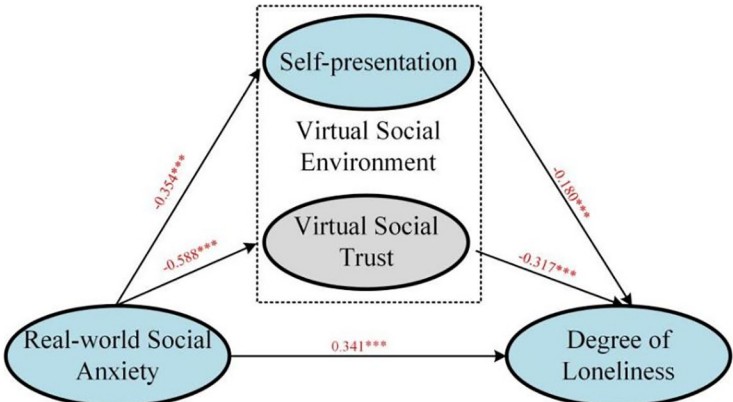

**Fig 2. Model diagram of virtual social behavior, real-world loneliness, and social anxiety.**

**Table 5. Bootstrap results (N = 368).**

| Route | Effect | Estimate | Bootstrap SE | Z | BC95% CI | | p | Mediation Effect Ratio |
|---|---|---|---|---|---|---|---|---|
| | | | | | MAX | MIN | | |
| A→ (B+C) →D | Total | 0.385 | 0.135 | 2.852 | 0.696 | 0.162 | 0.001 | 0.712 |
| | Direct | 0.111 | 0.147 | 0.755 | 0.402 | -0.174 | 0.451 | |
| | Indirect | 0.274 | 0.149 | 1.839 | 0.665 | 0.055 | 0.019 | |

Notes: A represents real-world social anxiety, B represents real self-presentation C represents virtual social trust, and D represents the degree of loneliness.

In this study, the bias-corrected Bootstrap testing method is used to analyze the total effect and mediation effect of virtual social behavior, real-world loneliness, and social anxiety. To determine the confidence interval of the mediation effect, we estimate the value to be approximately 2.5, with the 97.5th percentile being the primary reference point. The results will be presented in Table 5.

The results indicate (Table 5) a significant positive relationship between real-world social anxiety and the degree of loneliness (p<0.05). That self-presentation and virtual social trust act as mediators between real-world social anxiety and the degree of loneliness, accounting for approximately 71.2% of the mediation effect. According to the standards for testing mediation effects [49], a mediation effect is considered distinctive when the indirect effect Z exceeds 0.9 and P<0.05. However, if Z is lower than 1.96, the mediation effect is deemed incomplete. In the context of the structural equation model (SEM) analyzing virtual social behavior, real-world loneliness, and social anxiety, the observed Z value of 1.839 indicates an obvious incomplete mediation effect, suggesting the presence of other potential mediating variables.

**ii. Studies on the fourth, fifth and sixth hypotheses.** By conducting a mean of independent sample T-Test on matching pairs, this study examined the correlation between algorithm matching technologies and the degree of loneliness among users. The results, presented in Table 6, indicate that there is a significant difference in the extent of preferences for algorithm-matching users in terms of loneliness (p<0.001). Users who like algorithm matching have a higher degree of loneliness (M = 3.35) compared to those who dislike it (M = 2.964). However, the loneliness disparities between users who like algorithm matching (M = 3.35) and those who are indifferent (M = 3.378) are not statistically significant (p = 0.753>0.05). On the other hand, there was a significant difference in loneliness scores (p<0.001) between users who dislike algorithm matching and those who are indifferent. Users who dislike algorithm matching reported lower levels of loneliness in the real world, and this group differed significantly from the other types. Therefore, the result of H4 (Hypothesis 4) is negative.

By comparing the mean of independent sample T-Test in matching pairs, this study aims to explore the correlation between algorithm matching technologies and real self-presentation, based on the scale means of self-presentation and preferences on algorithm matching in virtual social settings. The results, as shown in Table 7, indicate that there is a significant difference in the extent of preferences on algorithm-matching users in terms of real self-presentation (p0.05). However, there are significant differences (p<0.001) in self-presentation between users who dislike algorithm matching and those who are indifferent. Users who dislike

**Table 6. Disparities in loneliness of user's preferences on algorithm matching.**

| Preferences | Like | | | Dislike | | | Indifference | | |
|---|---|---|---|---|---|---|---|---|---|
| | n | M | SD | n | M | SD | n | M | SD |
| Loneliness | 120 | 3.350 | 0.785 | 84 | 2.964 | 0.687 | 164 | 3.378 | 0.679 |

**Table 7. Disparities in self-presentation of user's preferences on algorithm matching.**

| Preferences | Like | | | Dislike | | | Indifference | | |
|---|---|---|---|---|---|---|---|---|---|
| | n | M | SD | n | M | SD | n | M | SD |
| Self-presentation | 120 | 2.667 | 0.927 | 84 | 3.222 | 0.891 | 164 | 2.703 | 0.713 |

algorithm matching prefer to show their real self-presentation on virtual social platforms, and their data also significantly differs from the other groups.

By comparing the mean of independent sample T-Test in matching pairs, this study aims to determine the correlation between algorithm matching technologies and virtual social trust. The results, shown in Table 8, indicate that there are slight disparities in real self-presentation based on the extent of preferences on algorithm-matching users (p = 0.059>0.05). On one hand, users who like algorithm matching have a mean real self-presentation score of 2.68, while those who dislike it have a mean score of 2.933. On the other hand, there are also slight disparities in real self-presentation (p = 0.157>0.05) between users who like algorithm matching (M = 2.68) and those who are indifferent (M = 2.522). However, the disparities in self-presentation between users disliking algorithm matching and those who are indifferent are significant (p<0.001). Users who dislike algorithm matching tend to show more real self-presentation on virtual social platforms, and this data differs significantly from the other categories. The relationship between real self-presentation and virtual social trust is positively interrelated, as confirmed in Table 3 (r = 0.658, p<0.01) and repeatedly supported by H5. In summary, H5 suggests that users who like algorithm matching do not necessarily show more authentic self-presentation or have more virtual social trust.

By conducting a mean comparison using independent sample T-Test on matching pairs, this study aimed to explore the correlation between algorithm matching technologies and real social anxiety, based on the scale means of virtual social trust and preferences on algorithm matching in virtual social settings. The results, presented in Table 9, indicate significant differences in the extent of preferences among algorithm-matching users in terms of real self-presentation (p<0.001). Users who like algorithm matching (M = 3.387) exhibit higher levels of real-world social anxiety compared to those who dislike it (M = 2.819). Additionally, significant disparities in real-world social anxiety are observed between users who like algorithm matching (M = 3.387) and those who are indifferent (M = 3.083) (p = 0.013<0.05). Similarly, there are noticeable differences in real-world social anxiety between users who dislike algorithm matching and those who are indifferent (p = 0.025<0.05). These findings suggest that users who prefer algorithm matching experience greater anxiety in real social interactions, and also support hypothesis H6.

## Part Five: Conclusions

### I. Virtual social exacerbates the degree of loneliness

The results showed that virtual social interactions actually increased feelings of loneliness. Additionally, real-world social anxiety had a direct impact on loneliness to some extent. Interestingly, there were correlations between real social anxiety, virtual social trust (real self-presentation), and loneliness. Users often turned to virtual social interactions to alleviate their loneliness caused by real social activities. However, due to negative experiences in real social situations, virtual social users tended to present themselves as more unreal and have less trust, which exacerbated their loneliness. Sherry's viewpoints were confirmed and it was found that lonelier users in real society tended to be more active on virtual social platforms. It was not the virtual social activity itself that amplified feelings of loneliness, but rather the trust and attitude

**Table 8. Disparities in virtual social trust of user's preferences on algorithm matching.**

| Preferences | Like | | | Dislike | | | Indifference | | |
|---|---|---|---|---|---|---|---|---|---|
| | n | M | SD | n | M | SD | n | M | SD |
| Virtual Social Trust | 120 | 2.680 | 0.988 | 84 | 2.933 | 0.859 | 164 | 2.522 | 0.838 |

towards virtual social interaction. The findings demonstrated that unreal self-presentation and mistrust increased feelings of loneliness in virtual social interaction. In this context, mutual trust plays a crucial role in determining the psychological effects of virtual social interaction [50]. Genuine social trust is what reduces loneliness in virtual social interactions. Research also shows that individuals with higher levels of loneliness are hesitant and fearful of self-presenting, lack trust in others, and carry real-life frustrations and self-humiliation into virtual social situations. This negative attitude towards real social situations affects virtual social interactions, leading to decreased mutual trust and increased concerns about negative comments, ultimately exacerbating social anxiety. Consequently, these users also fear negative comments on virtual social activities and experience conflicts with the identity and amity of other virtual social users, further intensifying their feelings of loneliness.

## II. Inefficient matching: Camouflages lead to inevitable loneliness

Regarding Hypotheses 4, 5, and 6, the main conclusion is that users who experience higher levels of loneliness and real social anxiety tend to engage in unreal self-presentation and exhibit lower levels of virtual social trust when utilizing the algorithm matching feature on the Soul App. These findings are not consistent with the intended purpose of the Soul App.

Individuals with higher levels of real social anxiety are not only motivated to make a positive impression on others but also care about their social skills [51]. Consequently, individuals who experience anxiety in online social settings tend to focus more on maintaining their self-image and self-improvement, which can lead to a reduced level of trust [52]. Moreover, individuals with higher levels of real social anxiety often have a negative perception of their own image, characteristics, and behaviors, and may lack self-identity. As a result, they tend to be pessimistic about sharing their personal information with society through online platforms. These individuals with lower self-awareness are particularly susceptible to the influence of negative online social comments [53]. Furthermore, individuals with more negative attitudes towards social anxiety are more prone to self-suspicion. Continuous experiences of rejection in social interactions can significantly impact their self-worth and result in low self-esteem [54]. On the other hand, some users strive to create an ideal image through self-disclosure, but the information they share can be a mix of authentic and fabricated messages. Individuals with weaker social abilities in online social contexts may struggle to differentiate between reality and the performance they present online [55]. Additionally, the degree of loneliness is found to be negatively associated with the self-presentation of online chatting. Users who are dissatisfied with online social experiences tend to feel more lonely [56]. The inconsistent social activities that are based on a fake or camouflaged self-image indicate a level of loneliness and social anxiety. As a result, these users find it difficult to select a partner who is algorithmically

**Table 9. Disparities in real-world social anxiety of user's preferences on algorithm matching.**

| Preferences | Like | | | Dislike | | | Indifference | | |
|---|---|---|---|---|---|---|---|---|---|
| | n | M | SD | n | M | SD | n | M | SD |
| Real-world Social Anxiety | 120 | 3.387 | 1.1617 | 84 | 2.819 | 0.910 | 164 | 3.083 | 0.855 |

matched and shares commonalities, because these partners are those whom they are self-denial about. Virtual social users aim to find a perfectly compatible partner with whom they can engage in mutual interactions.

Additionally, considering a different viewpoint, can algorithm technologies-driven matching truly satisfy the actual demands of users? In this context, the initial step of algorithm-matching-driven online social platforms involves not just communication between users, but also the more common human-computer interactions. The significance of AI lies in the specific relationship between humans and computers, rather than the technology itself or the subjective imagination of humans [57]. AI possesses a distinct advantage in terms of objectivity and efficiency in data processing. However, when it comes to the inherently complex process of social interactions, which involve subjective preferences and intricate thinking, the existing algorithm matching technologies struggle to fully comprehend and accurately capture the exact preferences of social partners. Perhaps, with advancements in AI technologies, the potential lies in exploring the use of decoding technologies such as electrocorticogram to guide human-computer interactions. These advancements could potentially offer new insights into the possibilities of direct exchanges of ideas.

## III. To address the problems or compromise: Possibilities of future social for strangers

Regarding the aforementioned factors, virtual social users should overcome virtual social barriers and present their real selves, demonstrating trust in their partners. When Soul App users become tired of shaping their self-image, they can log off their account, which will then remove their list of friends. The Soul App system will then recommend new partners based on the provided information. In this way, the matching function acts as a matching game. It is important to note that algorithm matching technologies do not simply enable quick social matching for virtual social users, but they also struggle to capture the true essence of virtual social interactions. Communicating with strangers online requires caution and doubt, and building mutual trust is crucial for algorithm matching-driven virtual social activities. However, it is important to recognize that the emotional impact of face-to-face communication far surpasses that of virtual social interactions. Furthermore, there are also drawbacks to algorithm matching-driven virtual social activities. When virtual social users heavily rely on algorithm matching, they become accustomed to virtual standards and habits of interaction, which can complicate real social communication. Additionally, virtual social platforms can lead to the development of split personalities and a blurred self-definition, as online users have the ability to shape multiple self-images and engage with a wide range of personalities and perspectives [58]. When users willingly subject themselves to technology filtering, they may find it increasingly difficult to think independently and critically. With the increasing reliance of online users on algorithm matching technologies, the fragmented mindset formed by various scattered information will dominate the thinking pattern of users, resembling the concept of a one-dimensional man as defined by Herbert Marcuse.

To address the limitations of virtual social interactions among strangers, it is advisable to create a combination of online and offline social activities that allow users to present their real selves. Additionally, providing guidance on socializing in different situations can foster closer relationships among a wider range of users. Tailored cooperation can strengthen social bonds, as trust built through collaboration supports the development of interpersonal connections. A key factor in successfully engaging in real social interactions with strangers is proximity. Being physically close or sharing common localities makes it easier for strangers to engage in real social activities. The advancements in 5G technology, coupled with quick and accurate

Navigation Positioning Systems, facilitate the development of distance-matching technologies that efficiently connect individuals. Within this technological context, it becomes feasible to create platforms that facilitate vertical socializing with strangers, allowing for more authentic scenarios. This approach effectively reduces anxiety associated with face-to-face interactions and mitigates feelings of loneliness.

## IV. Conclusion summary and study limitations

This study highlights the complex relationship between virtual social interactions, algorithm matching technologies, and users' psychological states, such as loneliness and social anxiety. It was found that virtual social interactions, rather than alleviating loneliness, often exacerbate it, particularly when users engage in unreal self-presentation and lack trust in virtual partners. The findings suggest that the effectiveness of algorithm matching is limited by its inability to accurately capture users' true preferences, leading to increased feelings of loneliness. Additionally, social anxiety in real-world contexts significantly affects users' behavior in virtual environments, where a lack of trust and self-doubt further contribute to negative social experiences. To mitigate these issues, the research proposes a future direction that combines both online and offline social activities, fostering real-life interactions and trust-building. Advances in AI and technologies like 5G and Navigation Positioning Systems could potentially offer new solutions to enhance social interactions, reduce anxiety, and address loneliness more effectively.

One key limitation of this study is the generalizability of the findings. The sample is drawn exclusively from users of the SoulApp, which caters to a specific demographic of individuals who seek virtual social interactions with strangers. This demographic may not represent the broader population that uses other virtual social platforms or engages in online interactions in different contexts. As such, caution should be exercised when applying these findings to general social media users. Future research could expand the scope by including users from various platforms and considering more diverse demographic variables to enhance the representativeness of the sample.

## Supporting information

**S1 File. The relevant data file for this article.**
(XLSX)

## Author Contributions

**Conceptualization:** Linsha Liu.

**Data curation:** Linsha Liu.

**Formal analysis:** Linsha Liu.

**Investigation:** Linsha Liu.

**Methodology:** Linsha Liu.

**Writing – original draft:** Linsha Liu.

**Writing – review & editing:** Linsha Liu.

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
