## [Decision Letter · Decision Letter 0]

9 Sep 2024

PONE-D-24-18667Research on the Relationship between Virtual Social Interaction and the Degree of Loneliness Based on Algorithm Matching Technologies: A Quantitative Analysis on the SOUL APP-A Virtual Social Software for StrangersPLOS ONE

Dear Dr. Liu,

Thank you for submitting your manuscript to PLOS ONE. After careful consideration, we feel that it has merit but does not fully meet PLOS ONE’s publication criteria as it currently stands. Therefore, we invite you to submit a revised version of the manuscript that addresses the points raised during the review process.

The study explores an interesting topic on social interaction in virtual social platforms and participants' emotions. The paper is well-structured, and the data analysis is sound. However, there are a few concerns raised by the reviewers. Please address these concerns and resubmit your revised manuscript.

**My comments:**

**This study investigates the relationship between social emotions, such as loneliness, and interpersonal communication on virtual social platforms using survey data. The conclusions are well-supported by the data, and the paper is well-organized. Therefore, I recommend it for publication. However, I have two suggestions:**

**It would be beneficial to include a separate paragraph summarizing the study's conclusions, as the current sections on the conclusions are somewhat scattered.**

**A discussion on the limitations of the study, particularly regarding the generalizability of the sample, would improve the paper.**

We look forward to receiving your revised manuscript.

Kind regards,

Yunhe Tong

Academic Editor

PLOS ONE

**Journal Requirements:**

**Additional Editor Comments:**

This study investigates the relationship between social emotions, such as loneliness, and interpersonal communication on virtual social platforms using survey data. The conclusions are well-supported by the data, and the paper is well-organized. Therefore, I recommend it for publication. However, I have two suggestions:

It would be beneficial to include a separate paragraph summarizing the study's conclusions, as the current sections on the conclusions are somewhat scattered and not focused.

A discussion on the limitations of the study, particularly regarding the generalizability of the sample, would improve the paper.

Reviewers' comments:

Reviewer's Responses to Questions

**Comments to the Author**

1. Is the manuscript technically sound, and do the data support the conclusions?

Reviewer #1: Yes

2. Has the statistical analysis been performed appropriately and rigorously? 

Reviewer #1: Yes

3. Have the authors made all data underlying the findings in their manuscript fully available?

Reviewer #1: No

4. Is the manuscript presented in an intelligible fashion and written in standard English?

Reviewer #1: No

5. **Review Comments to the Author**

Reviewer #1: Thank you for submitting your paper. Unfortunately, I recommend major revisions. On page 8, it is unclear what the opening passage is talking about. This happens again 9. I have attached a PDF containing further comments. I suggest the author address all of them before resubmitting.

6. PLOS authors have the option to publish the peer review history of their article (what does this mean?). If published, this will include your full peer review and any attached files.

Reviewer #1: No

---

## [Author Response · Author response to Decision Letter 0]

25 Sep 2024

Response Letter

Dear editors and reviewers,

 I am very grateful for your constructive comments and suggestions for my manuscript entitled" Research on the Relationship between Virtual Social Interaction and the Degree of Loneliness Based on Algorithm Matching Technologies: A Quantitative Analysis on the Soul App-A Virtual Social Software for Strangers"(ID: PONE-D-24-18667).Your comments are very valuable and helpful for improving my manuscript. In the following, the responses to all the comments are provided one by one.

 I have tried my best to make all the revisions clear, and I hope that the revised manuscript can satisfy the requirements for publication.

 The responses and revisions regarding the two suggestions made by the reviewer are as follows:

1.It would be beneficial to include a separate paragraph summarizing the study's conclusions, as the current sections on the conclusions are somewhat scattered.

 The added summary of the conclusions: 

 This study highlights the complex relationship between virtual social interactions, algorithm matching technologies, and users' psychological states, such as loneliness and social anxiety. It was found that virtual social interactions, rather than alleviating loneliness, often exacerbate it, particularly when users engage in unreal self-presentation and lack trust in virtual partners. The findings suggest that the effectiveness of algorithm matching is limited by its inability to accurately capture users' true preferences, leading to increased feelings of loneliness. Additionally, social anxiety in real-world contexts significantly affects users' behavior in virtual environments, where a lack of trust and self-doubt further contribute to negative social experiences. To mitigate these issues, the research proposes a future direction that combines both online and offline social activities, fostering real-life interactions and trust-building. Advances in AI and technologies like 5G and Navigation Positioning Systems could potentially offer new solutions to enhance social interactions, reduce anxiety, and address loneliness more effectively.

2.A discussion on the limitations of the study, particularly regarding the generalizability of the sample, would improve the paper.

The added discussion of the study limitations: 

One key limitation of this study is the generalizability of the findings. The sample is drawn exclusively from users of the SoulApp, which caters to a specific demographic of individuals who seek virtual social interactions with strangers. This demographic may not represent the broader population that uses other virtual social platforms or engages in online interactions in different contexts. As such, caution should be exercised when applying these findings to general social media users. Future research could expand the scope by including users from various platforms and considering more diverse demographic variables to enhance the representativeness of the sample

 The following points 1 to 17 address the various issues and comments noted by the reviewer in the text, with corresponding revisions and responses.

1.The second sentence of the first paragraph (lines23-26) is unclear and has been revised as follows：

“Through the application of big data and intelligent technologies, human cognitive processes, motor skills, and even cultural preferences are increasingly being intelligently matched and synchronized. As a result, audiences gain a stronger sense of immersion and a more intuitive online experience.”

2.To maintain consistency, the term "Soul" has been used uniformly throughout the text, replacing all instances of the all-uppercase "SOUL."

3.The second paragraph of the introduction has been completely rewritten to clearly articulate the research objectives and significance of this study.The revision is as follows(lines 32-40):

“This study takes the Soul App as a case to explore the impact of AI-driven virtual social behaviors on individual loneliness. It aims to uncover how the authenticity of self-presentation in virtual interactions, the sense of trust in these environments, the pressures of real-world social interactions, and loneliness interact to create a cyclical effect. Additionally, the study seeks to evaluate the effectiveness of AI matching technology in enhancing authenticity and trust in virtual social interactions, and whether these improvements can alleviate users' feelings of loneliness. The study ultimately aims to provide insights into the effects of virtual social interactions on individual mental health, offering valuable guidance for the development and design of virtual technologies.”

4.The first paragraph of the research background was not very clear, so I rewrote the entire section to better explain the logical starting point of the research.The revision is as follows(lines 42-63):

"Through technology, we manifest and extend our thoughts, injecting them into the objective world and shaping it according to our designs," Paul Levinson（1988） emphasizes human agency. He argues that in the evolution of media, rational choices made by humans can address both the shortcomings in media development and the limitations inherent in media themselves. Media technologies, as they evolve, increasingly tend to meet human needs and facilitate the exchange of information (Jiang Xiaoli & Jia Ruiqi, 2017). Each new medium is a remedy for its predecessors, making media more and more suited to human needs. "Why do we expect more from technology, yet cannot become closer to one another?" This is the question posed by Sherry Turkle（2014）, a professor of sociology at MIT, about the influence of internet technology on human interactions in the real world. Supported by network technology, people can maintain constant contact with others beyond the constraints of time and space, yet at the same time, they ignore those who are physically near. With the development of network technology, people seem to have become more vulnerable; the more noise there is in the virtual world, the more loneliness they experience in reality. Sherry Turkle's views have resonated widely with the public, and the term 'Alone Together' has become a reflection of the psychological state of people in the networked society. Both Levinson and Turkle believe that the development of media technology has a significant impact on humanity, yet their conclusions are completely opposite. Levinson argues that the evolution of media adapts to human needs, while Turkle believes that technological development inevitably leads to human alienation. In the digital age, how does the issue of 'Alone Together' differ from what Sherry Turkle observed in her 2014 study? As technology achieves higher levels of development, does new media technology accommodate human needs, or does it further alienate humanity?

5.The definition of loneliness and related research have been further clarified(The second paragraph of the first subsection in the part two, Lines 89-93):

Loneliness, on the other hand, refers to the experience of emotional and social isolation, highlighting the subjective nature of this feeling(Weiss, 1975). Russell et. al. (1978) developed a measure of loneliness to quantify this subjective experience, emphasizing the need for standardized tools to assess loneliness. Moore et. al.(1983) focused on loneliness in adolescence, examining its correlates, attributions, and coping mechanisms among young individuals.

6.The impact of social anxiety on loneliness was re-examined, and relevant literature was supplemented(Lines 94-106).

Numerous studies have shown a strong positive correlation between loneliness and social anxiety in both youth and adulthood (Spindler et al., 2007). Research by Sun Mengyuan and Liu Kun (2018) demonstrated that social anxiety significantly predicts loneliness in middle school students. In older adults, Sun et al. (2024) further emphasized that social anxiety strongly predicts loneliness, with perceived social support moderating this effect. This establishes the foundation for exploring the relationship between loneliness in virtual social interactions and real-world social anxiety, indicating that anxiety and loneliness are mutually reinforcing.Additionally, Enez et al. (2016) found a positive correlation between smartphone addiction, social anxiety, and loneliness among university students in Istanbul. Zhou et al. (2024), using a network approach, revealed that loneliness and escapism mediate the relationship between social anxiety and social comfort. Based on this, we deduce that frequent use of smartphones or virtual social platforms may exacerbate loneliness, especially when users are trying to escape real-world social pressures. Therefore, we propose the first hypothesis:

7.The literature review on the relationship between self-presentation and social anxiety has been rewritten to more clearly propose hypotheses(Lines 124-146).

The association between self-presentation and social anxiety has been a focal point in numerous research studies. Maddux et. al.(1988)explored the combination of self-efficacy theory and self-presentation theory in understanding social anxiety, with a focus on the connection between general social anxiousness and expectations regarding anxiety-inducing social situations. Martin et. al.(1996) emphasized the significance of physical self-presentation in the context of trait anxiety during sports competitions, highlighting the relationship between fear of negative evaluation and competitive trait anxiety. Gammage et. al.(2004) investigated the cognitive aspects of self-presentation in exercise settings, examining variables such as social physique anxiety, self-presentational efficacy, impression motivation, and exercise imagery among female exercisers. Moreover, investigations conducted by Jankauskienė et al.(2018) analyzed the association between social anxiety and problematic internet use, exercise adherence, body image, and social physique anxiety. These studies highlight the influence of self-presentation on individuals' behaviors and psychological well-being in various contexts. Recent research by Kehayes et al.(2019) delved into the connection between perfectionistic self-presentation and social anxiety, emphasizing the mediating roles of self-presentation motivation, expectancies, and anger rumination. These studies collectively indicate that self-presentation concerns have a substantial impact on behavior and anxiety levels in various social and competitive environments. 

The literature review on the association between self-presentation and social anxiety demonstrates a multifaceted link across various contexts such as social interactions, competitive settings, exercise environments, and online platforms. The studies discussed underscore the importance of considering self-presentation concerns in comprehending social anxiety and its implications for individuals' behaviors and psychological well-being.

8.The definition of algorithm matching was added at the beginning of Section 3 in Part 2(Lines 184-191).

Matching plays a vital role in the rational allocation of resources in many areas, ranging from market operation to people's daily lives. In economics, the term matching theory is coined for pairing two agents in a specific market to reach a stable or optimal state. In computer science, all branches of matching problems have emerged, such as the question-answer matching in information retrieval, user-item matching in a recommender system, and entity-relation matching in the knowledge graph(Ren J, et al., 2021). A matching algorithm is a type of algorithm used to dynamically compute similarity and synergy between different entities based on semantics and properties.(Trokanas N, et al., 2013). 

9.In Section 3 of Part 2, official content from the Soul App was added to explain its product concept of using algorithm matching to promote social interactions among users, providing a foundation for proposing further hypotheses(Lines 193-201).

For example, in the case of the Soul App, after registering, users take a "Soul Assessment" . They can also add "gravitational tags" to better showcase themselves. Using the results of the "Soul Assessment" and "gravitational tags," Soul's AI system analyzes users' interests and behavior to quickly recommend suitable, high-quality connections(Soul App Web, 2024). By utilizing artificial intelligence technology and targeted algorithms in this way to achieve personalized recommendations and improve the efficiency of matching relevant user information, this could create social interactions that better meet users' actual needs and reduce loneliness. Therefore, the fourth hypothesis is proposed:

10.The specific daily active user count for the Soul App has been clarified(Lines 244-245).

Soul App has 9.55 million daily active users

11.The incorrect survey date has been corrected(Lines 248).

Data collection for this study from Nov. 1, 2023 to Nov 20, 2023.

12.Clarified the platform for distributing the questionnaire(Lines 248-250).

The survey for this study was created using the online survey platform Wenjuanxing (www.wjx.cn, accessed on Nov. 1, 2023) and disseminated through Soul App.

13.Explained the reasons for adding another scale as a reference for the loneliness questionnaire(Lines 279-282).

Based on a simplified version of the Loneliness Scale (Hughes, M. E., et al., 2004) and referring to the mental health assessment scale developed by Wang Xu et al. (WANG X, et al., 1999), which is suitable for Chinese individuals (currently, there is no mature loneliness scale specifically based on the Chinese population)

14.Regarding users' preference for algorithm matching, this study employed simple question inquiries instead of scales. These questions have been specifically detailed in Section 4 Tests of Hypotheses of Part4. Therefore, the fifth point of Section 2 in Part 3 has been removed.

15.It further explains the representativeness of the questionnaire sample structure(Lines 310-315).

The sample characteristics collected through the questionnaire survey are basically consistent with the user characteristics displayed by the Soul App（Ryanben Capital,2023）, which indicates that the sample obtained from the questionnaire has good representativeness and can effectively reflect the actual characteristics of Soul App users. This also demonstrates the reliability and validity of the research findings, enhancing the credibility of the study.

16.The formatting errors in the text were corrected to address the issue of missing labels in the figures(Lines 376).

17.For the research on ii. Studies on the Fourth, Fifth, and Sixth Hypotheses, the measurement method involved conducting an independent samples T-Test on matching pairs. This study examined the correlation between algorithm matching technologies and the degree of loneliness, real self-presentation, virtual social trust, and real social anxiety, based on the scale means of loneliness, self-presentation, virtual social trust, anxiety in real-world social activities, and preferences for algorithm matching in virtual social settings. These measurement methods are explained in lines 397-398,lines 411-414,lines 422-423 and lines 439-442.

Sincerely

Corresponding Author.

Linsha Liu

---

## [Editor Report · Decision Letter 1]

2 Oct 2024

PONE-D-24-18667R1Research on the Relationship between Virtual Social Interaction and the Degree of Loneliness Based on Algorithm Matching Technologies: A Quantitative Analysis on the Soul App-A Virtual Social Software for StrangersPLOS ONE

Dear Dr. Liu,

Thank you for submitting your manuscript to PLOS ONE. After careful consideration, we feel that it has merit but does not fully meet PLOS ONE’s publication criteria as it currently stands. Therefore, we invite you to submit a revised version of the manuscript that addresses the points raised during the review process.

We look forward to receiving your revised manuscript.

Kind regards,

Yunhe Tong

Academic Editor

PLOS ONE

**Additional Editor Comments:**

The reviewer raised concerns about the unclear explanation in some sections, and requested further details, particularly regarding the methods and results. As a result, I have decided to proceed with a major revision.

---

## [Author Response · Author response to Decision Letter 1]

4 Oct 2024

Response Letter

Dear editors and reviewers,

 I am very grateful for your constructive comments and suggestions for my manuscript entitled" Research on the Relationship between Virtual Social Interaction and the Degree of Loneliness Based on Algorithm Matching Technologies: A Quantitative Analysis on the Soul App-A Virtual Social Software for Strangers"(ID: PONE-D-24-18667).Your comments are very valuable and helpful for improving my manuscript. In the following, the responses to all the comments are provided one by one.

 I have tried my best to make all the revisions clear, and I hope that the revised manuscript can satisfy the requirements for publication.

 The responses and revisions regarding the two suggestions made by the reviewer are as follows:

1.It would be beneficial to include a separate paragraph summarizing the study's conclusions, as the current sections on the conclusions are somewhat scattered.

 The added summary of the conclusions: 

 This study highlights the complex relationship between virtual social interactions, algorithm matching technologies, and users' psychological states, such as loneliness and social anxiety. It was found that virtual social interactions, rather than alleviating loneliness, often exacerbate it, particularly when users engage in unreal self-presentation and lack trust in virtual partners. The findings suggest that the effectiveness of algorithm matching is limited by its inability to accurately capture users' true preferences, leading to increased feelings of loneliness. Additionally, social anxiety in real-world contexts significantly affects users' behavior in virtual environments, where a lack of trust and self-doubt further contribute to negative social experiences. To mitigate these issues, the research proposes a future direction that combines both online and offline social activities, fostering real-life interactions and trust-building. Advances in AI and technologies like 5G and Navigation Positioning Systems could potentially offer new solutions to enhance social interactions, reduce anxiety, and address loneliness more effectively.

2.A discussion on the limitations of the study, particularly regarding the generalizability of the sample, would improve the paper.

The added discussion of the study limitations: 

One key limitation of this study is the generalizability of the findings. The sample is drawn exclusively from users of the SoulApp, which caters to a specific demographic of individuals who seek virtual social interactions with strangers. This demographic may not represent the broader population that uses other virtual social platforms or engages in online interactions in different contexts. As such, caution should be exercised when applying these findings to general social media users. Future research could expand the scope by including users from various platforms and considering more diverse demographic variables to enhance the representativeness of the sample

 The following points 1 to 17 address the various issues and comments noted by the reviewer in the text, with corresponding revisions and responses.

1.The second sentence of the first paragraph (lines23-26) is unclear and has been revised as follows：

“Through the application of big data and intelligent technologies, human cognitive processes, motor skills, and even cultural preferences are increasingly being intelligently matched and synchronized. As a result, audiences gain a stronger sense of immersion and a more intuitive online experience.”

2.To maintain consistency, the term "Soul" has been used uniformly throughout the text, replacing all instances of the all-uppercase "SOUL."

3.The second paragraph of the introduction has been completely rewritten to clearly articulate the research objectives and significance of this study.The revision is as follows(lines 32-40):

“This study takes the Soul App as a case to explore the impact of AI-driven virtual social behaviors on individual loneliness. It aims to uncover how the authenticity of self-presentation in virtual interactions, the sense of trust in these environments, the pressures of real-world social interactions, and loneliness interact to create a cyclical effect. Additionally, the study seeks to evaluate the effectiveness of AI matching technology in enhancing authenticity and trust in virtual social interactions, and whether these improvements can alleviate users' feelings of loneliness. The study ultimately aims to provide insights into the effects of virtual social interactions on individual mental health, offering valuable guidance for the development and design of virtual technologies.”

4.The first paragraph of the research background was not very clear, so I rewrote the entire section to better explain the logical starting point of the research.The revision is as follows(lines 42-63):

"Through technology, we manifest and extend our thoughts, injecting them into the objective world and shaping it according to our designs," Paul Levinson（1988） emphasizes human agency. He argues that in the evolution of media, rational choices made by humans can address both the shortcomings in media development and the limitations inherent in media themselves. Media technologies, as they evolve, increasingly tend to meet human needs and facilitate the exchange of information (Jiang Xiaoli & Jia Ruiqi, 2017). Each new medium is a remedy for its predecessors, making media more and more suited to human needs. "Why do we expect more from technology, yet cannot become closer to one another?" This is the question posed by Sherry Turkle（2014）, a professor of sociology at MIT, about the influence of internet technology on human interactions in the real world. Supported by network technology, people can maintain constant contact with others beyond the constraints of time and space, yet at the same time, they ignore those who are physically near. With the development of network technology, people seem to have become more vulnerable; the more noise there is in the virtual world, the more loneliness they experience in reality. Sherry Turkle's views have resonated widely with the public, and the term 'Alone Together' has become a reflection of the psychological state of people in the networked society. Both Levinson and Turkle believe that the development of media technology has a significant impact on humanity, yet their conclusions are completely opposite. Levinson argues that the evolution of media adapts to human needs, while Turkle believes that technological development inevitably leads to human alienation. In the digital age, how does the issue of 'Alone Together' differ from what Sherry Turkle observed in her 2014 study? As technology achieves higher levels of development, does new media technology accommodate human needs, or does it further alienate humanity?

5.The definition of loneliness and related research have been further clarified(The second paragraph of the first subsection in the part two, Lines 89-93):

Loneliness, on the other hand, refers to the experience of emotional and social isolation, highlighting the subjective nature of this feeling(Weiss, 1975). Russell et. al. (1978) developed a measure of loneliness to quantify this subjective experience, emphasizing the need for standardized tools to assess loneliness. Moore et. al.(1983) focused on loneliness in adolescence, examining its correlates, attributions, and coping mechanisms among young individuals.

6.The impact of social anxiety on loneliness was re-examined, and relevant literature was supplemented(Lines 94-106).

Numerous studies have shown a strong positive correlation between loneliness and social anxiety in both youth and adulthood (Spindler et al., 2007). Research by Sun Mengyuan and Liu Kun (2018) demonstrated that social anxiety significantly predicts loneliness in middle school students. In older adults, Sun et al. (2024) further emphasized that social anxiety strongly predicts loneliness, with perceived social support moderating this effect. This establishes the foundation for exploring the relationship between loneliness in virtual social interactions and real-world social anxiety, indicating that anxiety and loneliness are mutually reinforcing.Additionally, Enez et al. (2016) found a positive correlation between smartphone addiction, social anxiety, and loneliness among university students in Istanbul. Zhou et al. (2024), using a network approach, revealed that loneliness and escapism mediate the relationship between social anxiety and social comfort. Based on this, we deduce that frequent use of smartphones or virtual social platforms may exacerbate loneliness, especially when users are trying to escape real-world social pressures. Therefore, we propose the first hypothesis:

7.The literature review on the relationship between self-presentation and social anxiety has been rewritten to more clearly propose hypotheses(Lines 124-146).

The association between self-presentation and social anxiety has been a focal point in numerous research studies. Maddux et. al.(1988)explored the combination of self-efficacy theory and self-presentation theory in understanding social anxiety, with a focus on the connection between general social anxiousness and expectations regarding anxiety-inducing social situations. Martin et. al.(1996) emphasized the significance of physical self-presentation in the context of trait anxiety during sports competitions, highlighting the relationship between fear of negative evaluation and competitive trait anxiety. Gammage et. al.(2004) investigated the cognitive aspects of self-presentation in exercise settings, examining variables such as social physique anxiety, self-presentational efficacy, impression motivation, and exercise imagery among female exercisers. Moreover, investigations conducted by Jankauskienė et al.(2018) analyzed the association between social anxiety and problematic internet use, exercise adherence, body image, and social physique anxiety. These studies highlight the influence of self-presentation on individuals' behaviors and psychological well-being in various contexts. Recent research by Kehayes et al.(2019) delved into the connection between perfectionistic self-presentation and social anxiety, emphasizing the mediating roles of self-presentation motivation, expectancies, and anger rumination. These studies collectively indicate that self-presentation concerns have a substantial impact on behavior and anxiety levels in various social and competitive environments. 

The literature review on the association between self-presentation and social anxiety demonstrates a multifaceted link across various contexts such as social interactions, competitive settings, exercise environments, and online platforms. The studies discussed underscore the importance of considering self-presentation concerns in comprehending social anxiety and its implications for individuals' behaviors and psychological well-being.

8.The definition of algorithm matching was added at the beginning of Section 3 in Part 2(Lines 184-191).

Matching plays a vital role in the rational allocation of resources in many areas, ranging from market operation to people's daily lives. In economics, the term matching theory is coined for pairing two agents in a specific market to reach a stable or optimal state. In computer science, all branches of matching problems have emerged, such as the question-answer matching in information retrieval, user-item matching in a recommender system, and entity-relation matching in the knowledge graph(Ren J, et al., 2021). A matching algorithm is a type of algorithm used to dynamically compute similarity and synergy between different entities based on semantics and properties.(Trokanas N, et al., 2013). 

9.In Section 3 of Part 2, official content from the Soul App was added to explain its product concept of using algorithm matching to promote social interactions among users, providing a foundation for proposing further hypotheses(Lines 193-201).

For example, in the case of the Soul App, after registering, users take a "Soul Assessment" . They can also add "gravitational tags" to better showcase themselves. Using the results of the "Soul Assessment" and "gravitational tags," Soul's AI system analyzes users' interests and behavior to quickly recommend suitable, high-quality connections(Soul App Web, 2024). By utilizing artificial intelligence technology and targeted algorithms in this way to achieve personalized recommendations and improve the efficiency of matching relevant user information, this could create social interactions that better meet users' actual needs and reduce loneliness. Therefore, the fourth hypothesis is proposed:

10.The specific daily active user count for the Soul App has been clarified(Lines 244-245).

Soul App has 9.55 million daily active users

11.The incorrect survey date has been corrected(Lines 248).

Data collection for this study from Nov. 1, 2023 to Nov 20, 2023.

12.Clarified the platform for distributing the questionnaire(Lines 248-250).

The survey for this study was created using the online survey platform Wenjuanxing (www.wjx.cn, accessed on Nov. 1, 2023) and disseminated through Soul App.

13.Explained the reasons for adding another scale as a reference for the loneliness questionnaire(Lines 279-282).

Based on a simplified version of the Loneliness Scale (Hughes, M. E., et al., 2004) and referring to the mental health assessment scale developed by Wang Xu et al. (WANG X, et al., 1999), which is suitable for Chinese individuals (currently, there is no mature loneliness scale specifically based on the Chinese population)

14.Regarding users' preference for algorithm matching, this study employed simple question inquiries instead of scales. These questions have been specifically detailed in Section 4 Tests of Hypotheses of Part4. Therefore, the fifth point of Section 2 in Part 3 has been removed.

15.It further explains the representativeness of the questionnaire sample structure(Lines 310-315).

The sample characteristics collected through the questionnaire survey are basically consistent with the user characteristics displayed by the Soul App（Ryanben Capital,2023）, which indicates that the sample obtained from the questionnaire has good representativeness and can effectively reflect the actual characteristics of Soul App users. This also demonstrates the reliability and validity of the research findings, enhancing the credibility of the study.

16.The formatting errors in the text were corrected to address the issue of missing labels in the figures(Lines 376).

17.For the research on ii. Studies on the Fourth, Fifth, and Sixth Hypotheses, the measurement method involved conducting an independent samples T-Test on matching pairs. This study examined the correlation between algorithm matching technologies and the degree of loneliness, real self-presentation, virtual social trust, and real social anxiety, based on the scale means of loneliness, self-presentation, virtual social trust, anxiety in real-world social activities, and preferences for algorithm matching in virtual social settings. These measurement methods are explained in lines 397-398,lines 411-414,lines 422-423 and lines 439-442.

Sincerely

Corresponding Author.

Linsha Liu

---

## [Editor Report · Decision Letter 2]

9 Oct 2024

Research on the Relationship between Virtual Social Interaction and the Degree of Loneliness Based on Algorithm Matching Technologies: A Quantitative Analysis on the Soul App-A Virtual Social Software for Strangers

PONE-D-24-18667R2

Dear Dr. Liu,

We’re pleased to inform you that your manuscript has been judged scientifically suitable for publication and will be formally accepted for publication once it meets all outstanding technical requirements.

Kind regards,

Yunhe Tong

Academic Editor

PLOS ONE
---

## [Editor Report · Acceptance letter]

25 Oct 2024

PONE-D-24-18667R2 

PLOS ONE

Dear Dr. Liu, 

I'm pleased to inform you that your manuscript has been deemed suitable for publication in PLOS ONE. Congratulations! Your manuscript is now being handed over to our production team.

Kind regards, 

on behalf of

Dr. Yunhe Tong 

Academic Editor

PLOS ONE